

# Sylvatic host associations of Triatominae and implications for Chagas disease reservoirs: a review and new host records based on archival specimens

Anna Y. Georgieva[*], Eric R.L. Gordon[*] and Christiane Weirauch

Department of Entomology, University of California, Riverside, CA, United States of America
[*] These authors contributed equally to this work.

Corresponding author
Eric R.L. Gordon, egord003@ucr.edu, erg55@cornell.edu

## ABSTRACT

**Background**. The 152 extant species of kissing bug include important vectors of the debilitating, chronic, and often fatal Chagas disease, which affects several million people mainly in Central and South America. An understanding of the natural hosts of this speciose group of blood-feeding insects has and will continue to aid ongoing efforts to impede the spread of Chagas disease. However, information on kissing bug biology is piecemeal and scattered, developed using methods with varying levels of accuracy over more than 100 years. Existing host records are heavily biased towards well-studied primary vector species and are derived from primarily three different types of observations, associational, immunological or DNA-based, with varying reliability.
**Methods**. We gather a comprehensive and unparalleled number of sources reporting host associations via rigorous targeted searches of publication databases to review all known natural, or sylvatic, host records including information on how each record was collected. We integrate this information with novel host records obtained via attempted amplification and sequencing of a ∼160 base pair (bp) region of the vertebrate 12S mitochondrial gene from the gastrointestinal tract of 64 archival specimens of Triatominae representing 19 species collected primarily in sylvatic habitats throughout the southern United States and Central and South America during the past 10 years. We show the utility of this method for uncovering novel and under-studied groups of Triatominae hosts, as well as detecting the presence of the Chagas disease pathogen via Polymerase Chain Reaction (PCR) of a ∼400 bp sequence of the trypanosome 18S gene.
**Results**. New host associations for several groups of arboreal mammals were determined including sloths, New World monkeys, coatis, arboreal porcupines and, for the first time as a host of any Triatominae, tayras. A thorough review of previously documented sylvatic hosts, organized by triatomine species and the type of observation (associational, antibody-based, or DNA-based), is presented in a phylogenetic context and highlights large gaps in our knowledge of Triatominae biology.
**Conclusion**. The application of DNA-based methods of host identification towards additional species of Triatominae, including rarely collected species that may require use of archival specimens, is the most efficient and promising way to resolve recognized shortfalls.

## INTRODUCTION

Triatominae, or kissing bugs, are blood-feeding members of the primarily predatory insect family Reduviidae (Order Hemiptera). This subfamily consists of 152 living species, nearly all distributed in the Americas, though several species occur in the Oriental region and *Triatoma rubrofasciata* (De Geer) is considered invasive throughout the tropics (*Otálora-Luna et al., 2015*; *Justi et al., 2014*). These insects are vectors of *Trypanosoma cruzi* Chagas, the parasite responsible for Chagas disease. Chagas disease is a chronic debilitating disease, prevalent in Latin America, and affecting up to 10 million people worldwide (*Pereira & Navarro, 2013*). There is no vaccine or effective cure once the symptoms of the chronic disease have manifested and the disease has been termed an emerging threat of the 21st century (*Bonney, 2014*; *Viotti et al., 2014*).

Most kissing bug species are suspected to be oligo- or polyphagous across a broad range of wild mammal and other vertebrate species (*Lent & Wygodzinsky, 1979*; *Galvão & Justi, 2015*). Many Triatominae hosts are sylvatic mammals, but domestic mammals such as dogs, cats, and rodents can also be fed upon and act as reservoir hosts of the parasite (*Deane, 1964*). Generalist kissing bug species will also feed on humans and some generalist species, e.g., *Triatoma infestans* Klug and *Rhodnius prolixus* (Stål), can exhibit infection rates with *T. cruzi* equal to or even higher than 40% (*Pizarro & Stevens, 2008*; *Feliciangeli et al., 2002*; *Guhl, Pinto & Aguilera, 2009*) though *T. cruzi* infection rates vary considerably among populations. Some triatomine species will target certain hosts if available and avoid other potential blood meal sources (*Otálora-Luna et al., 2015*). *Cavernicola pilosa* Barber appears to only feed on bats (*Usinger, 1944*; *Oliveira et al., 2008*), while *Triatoma delpontei* Romaña and Abalos and *Psammolestes* Bergroth species are usually found in association with various birds (*Usinger, 1944*; *Salvatella et al., 1992*). In addition, there are reports of some kissing bug species feeding on other arthropods (*Garrouste, 2009*; *Sandoval et al., 2010*; *Kjos et al., 2013*), feeding on other engorged kissing bug individuals (*Sandoval et al., 2004*) and even facultative nectar feeding (*Díaz-Albiter et al., 2016*). The extent of these behaviors in a natural environment and for the great majority of kissing bug species is unknown. A diet lacking blood has been shown experimentally to result in complete or increased mortality in at least some species (*Durán, Siñani & Depickère, 2016*), or suggesting that arthropod feeding may be rare or driven by the lack of more suitable hosts. Overall, existing host association data is biased towards a handful of heavily studied and well-documented primary vector species and little data exists for many other Triatominae, particularly in sylvatic habitats (*Carcavallo, da Silva Rocha & Galindez Giron, 1998*). Understanding patterns of host associations across Triatominae can help to identify as yet underappreciated triatomine species of medical interest as well as determine populations of vertebrate hosts that have significant roles in sustaining vectors and potentially serve as reservoir hosts of *Trypanosoma cruzi*.

Records of Triatominae host associations are based on different types of observations that we here classify as associational (i.e., visual observation of a kissing bug in presumed or actual close association with a possible host), immunological, and DNA-based methods. Each of these approaches possess a unique set of advantages and disadvantages. Many early studies on kissing bug-host interactions depended on observations of the insects during laboratory feeding tests or, more rarely, in the wild (*Usinger, 1944*; *Packchanian, 1940*). Despite presenting direct evidence for feeding, laboratory experiments are necessarily unnatural. Laboratory tests either offer insects a single possible host species and observe whether feeding occurs (*Sant'Anna et al., 2001*; *Bodin, Vinauger & Lazzari, 2009*; *Martínez-Ibarra et al., 2007*) or present them with a choice between two or more hosts to determine preference (*Gürtler et al., 2009*). Both approaches may drive insects to feed on organisms that they would not feed upon in natural habitats. Observations of cohabitation of Triatominae with vertebrates in the wild, e.g., in animal nests and burrows, while more natural, are usually tentative because the association may not reflect actual feeding. There is likely also a bias towards more accessible terrestrial nests and burrows compared to corresponding arboreal habitats. Immunological methods to detect host associations in Triatominae were first used in the 1960s (*Barretto, 1967*; *Freitas, Siqueira & Ferreira, 1960*; *Pipkin, 1962*), seemingly overcoming problems posed by associational methods. By utilizing specific, but often polyclonal, antibodies in antisera developed for a predetermined range of potential host species, experimenters infer direct host associations. Precipitin tests have continued to be a popular way to detect host antigens in Triatominae blood meals (*Christensen & de Vasquez, 1981*; *Villela et al., 2010*; *Lorosa et al., 2003*). While a promising technique for forensic determination of actual hosts, disadvantages of precipitin tests are twofold. First, tests may suffer from non-specificity in antibody binding or irreproducibility as a result of variance in antibodies (*Baker, 2015*). Second, due to the cost of developing different sets of antibodies for different groups of hosts, antibodies are typically developed to be specific only for large groups of potential host vertebrates, such as rodents. This approach results in reduced resolution of hosts, i.e., vertebrates are not identified to genus or species, but will also fail to detect unexpected or rare host taxa for which no antibody set is available.

More recently, studies have begun to use DNA-based methods to detect host identity (*Kjos et al., 2013*; *Mota et al., 2007*; *Dias et al., 2010*). These methods typically use PCR that targets the conserved regions of variable, mitochondrial genes for amplification. When amplified sequences are compared to a known database, it is usually possible to determine which species or at least genus of organism the blood originated from. These studies have documented that certain species, such as *Triatoma rubida* (Uhler), *Triatoma protracta* (Uhler), *and Triatoma gerstaeckeri* (Stål), feed on a large variety of vertebrate hosts (*Kjos et al., 2013*; *Stevens et al., 2012*). PCR-based studies also have the potential to determine the percentage of specimens within a given kissing bug population that have fed on humans (*Stevens et al., 2012*). While PCR is useful in detecting a wide range of hosts, given that databases such as GenBank now hold a library of barcodes for most mammal species and many other vertebrates, it does have a relatively high risk for human contamination (*Lucero et al., 2014*). Primers can also have biases in amplifying DNA that closely mirrors their
sequence while not amplifying other sequences or also amplify the insect's own DNA, which can interfere with detecting host DNA from the blood sample. Multiple blood meals per specimen can amplify and interfere with determining a single sequence and must be separated via cloning of the PCR product or next generation sequencing.

While Chagas disease incidence has trended downward in the past thirty years due to screening of blood donations for *T. cruzi* as well as successful vector control applications of insecticides targeting domiciliated Triatominae species (*Moncayo & Silveira, 2009*), significant challenges remain for further reducing the spread of the disease. One remaining hurdle relates to sylvatic species of Triatominae that continue to transmit Chagas disease to humans primarily in rural areas (e.g., areas of the Amazon *Quinde-Calderón et al., 2016*). Studies focusing on the host associations of such species could help to inform policies to limit their impact, for example, by identifying primary vertebrate hosts of sylvatic species. However, most existing studies have focused on known primary vector species and often targeted only domestic or peridomestic habitats where the transmission risk to humans is considered higher (e.g., *Cecere et al., 2016*; *Cantillo-Barraza et al., 2015*). In addition, most previous DNA-based studies have surveyed narrow geographic areas, with several focusing on North America, and have used only living or very recently preserved specimens. While there are some aggregations of known hosts of Triatominae (e.g., *Carcavallo, da Silva Rocha & Galindez Giron, 1998*) as well as one analysis of host association patterns using combined precipitin data from 61 studies (*Rabinovich et al., 2011*) there exists no comprehensive review of reported Triatominae hosts. In particular, a synthesis of known triatomine-host associations that specifies the method through which the record was obtained and allows for assessing the reliability of the record is yet unavailable. Our study therefore has three objectives: (1) to contribute to the growing knowledge base of host associations across Triatominae, we conducted PCR of gastrointestinal contents extracted from 19 species of Triatominae from a range of localities (Bolivia to United States); most specimens were collected in sylvatic habitats using light traps and the sample comprises rarely encountered kissing bug species; (2) to determine the feasibility of assessing host associations and trypanosome infection for archival kissing bug specimens that have been preserved in ethanol for up to 10 years, we conducted PCR-based identification of host sequences and trypanosomes for 64 specimens; (3) to establish currently documented sylvatic host associations, we conducted a thorough literature review for all species of Triatominae while recording the method used to determine that association; host patterns and gaps in our current knowledge were visualized in phylogenetic context for both kissing bug species and vertebrate hosts.

## MATERIALS AND METHODS

### Taxon sampling

Triatominae specimens were primarily collected in sylvatic conditions via light trapping or hand collection throughout the southern United States and Central and South America and were preserved in ethanol (concentrations either unknown or 95%) between 2005 and 2015. We classify the habitat that each specimen as either domestic (found in a residence),

peridomestic (found outside in a residential area, near a residence) or sylvatic (found in natural habitats, sometimes near field research stations). It was our aim to survey as wide a variety of Triatominae species as possible, but certain species (e.g., *Triatoma protracta*, *Panstrongylus geniculatus* [Latreille]) were sampled more thoroughly due to the availability of specimens present in the Weirauch lab ethanol repository. One specimen of *Opisthacidius parkoi* (Lent and Wygodzinsky), a close predatory relative of Triatominae, was also sampled and served as a control. Voucher specimen data (unique specimen identifier [USI], determination, sex, specimen depository, collecting locality and event) were recorded using the Planetary Biodiversity Institute instance of the Arthropod Easy Capture database (research.amnh.org/pbi/locality). Images of voucher specimens were taken using a Leica DFC450 C Microsystems system with a Planapo 1.0x objective. Voucher data, including collecting technique and images, are available online at Heteroptera Species Pages (research.amnh.org/pbi/heteropteraspeciespage) and are best searched by species and then USI number.

## DNA extraction

DNA of the gastrointestinal contents of Triatominae specimens was extracted in order to perform PCR. To avoid cross-contamination as well as the recognized threat of contamination with human DNA, all equipment and work benches were sterilized (dissecting petri dish, forceps, iris scissors) before and after processing each specimen using 10% bleach. Cuticular surfaces of specimens were sterilized with 1% bleach for 3 min, to eliminate possible contaminants acquired before capture in ethanol or during ethanol storage. The thorax and abdomen were separated and contents of the abdomen were removed with forceps and placed into an Eppendorf tube. While performing this procedure, the sex of the specimen and whether blood was visible in the gut or not was recorded. When a large volume of blood was present, contents were divided into separate Eppendorf tubes. Gut contents were homogenized for 2 min with an Eppendorf pestle and DNA was extracted with a QIAGEN DNeasy blood and tissue kit. We recorded the amount of blood in each specimen on a scale of 1 to 4 where: 1—no material visible in digestive tract; 2—small amount of dark, digested blood present; 3—obvious blood present; 4—completely engorged with blood. For seven specimens of *Triatoma protracta* and one specimen of *Eratyrus mucronatus* Stål which had been extracted previously, we were not able to record the amount of blood present in the digestive tract.

## PCR

We tested seven previously developed sets of primers targeting three different mitochondrial genes (Table 1) designed for identification of vertebrate hosts from invertebrate blood meals via PCR on each of our extracts. We were not able to achieve consistent, acceptably broad or specific results for several sets of previously used primers listed in Table 1 (all those listed without asterisks). These primers amplified the corresponding DNA sequence of certain species of Triatominae or did not amplify DNA that should be present based on the results of other primer sets. We found that the "Kitano" 12S primers (*Kitano et al., 2007*) yielded the most consistent and the greatest number of amplified bands across samples among the

**Table 1  Primer sequences and PCR conditions used in this study.**

| DNA target | Primer set | Locus | Direction | Sequence | Annealing temperature | Reference |
|---|---|---|---|---|---|---|
| Vertebrate DNA | **Kitano 12S\*** | **12S** | **F** | 5′-**CCC AAA CTG GGA TTA GAT ACC C**-3′ | **57°** | *Alcaide et al. (2009)* |
| | | | **R** | 5′-**GTT TGC TGA AGA TGG CGG TA**-3′ | | *Alcaide et al. (2009)* |
| | Melton 12S | 12S | F | 5′-ACT GGG ATT AGA TAC CCC ACT ATG-3′ | 53° | *Hwang et al. (2010)* |
| | | | R | 5′-ATC GAT TAT AGA ACA GGC TCC TC-3′ | | *Hwang et al. (2010)* |
| | Vert COI | COI | M13BC-FW | 5′-TGT AAA ACG ACG GCC AGT HAA YCA YAA RGA YAT YGG NAC-3′ | 45° | *Kitano et al. (2007)* |
| | | | BCV-RV1 | 5′-GCY CAN AYY ATN CYY RTR TA-3′ | | *Kitano et al. (2007)* |
| | DC-CytB | CytB | UP | 5-CRT GAG GMC AAA TAT CHT TYT-3 | 42.5° | *Maslov et al. (1996)* |
| | | | DW | 5-ART ATC ATT CWG GTT TAA TRT-3 | | *Maslov et al. (1996)* |
| | Avian CytB | CytB | F | 5′-GAC TGT GAC AAA ATC CCN TTC CA-3′ | 55° | *Melton & Holland (2007)* |
| | | | R | 5′-GGT CTT CAT CTY HGG YTT ACA AGA C-3′ | | *Melton & Holland (2007)* |
| | Mammalian CytB | CytB | F | 5′-CGA AGC TTG ATA TGA AAA ACC ATC GTT G-3′ | 55° | *Melton & Holland (2007)* |
| | | | R | 5′-TGT AGT TRT CWG GGT CHC CTA-3′ | | *Melton & Holland (2007)* |
| | Vert CytB | CytB | CB1-L | 5′-CCC CTC AGA ATA TTT GTC CTC A-3′ | 57° | *Mota et al. (2007)* |
| | | | CB2-H | 5′-CAT CCA ACA TCT CAG CAT GAT GAA A-3′ | | *Mota et al. (2007)* |
| Trypanosome DNA | **Tcz\*** | **18S** | **18sf** | 5′-**TTA ACG GGA ATA TCC TCA GC**-3′ | **50°** | *Ngo & Kramer (2003)* |
| | | | **S829r** | 5′-**GCA TCA CAG ACC TGC TGT TG**-3′ | | *Palumbi et al. (1991)* |

primer sets tested and this is the only primer set for which we present results. If samples yielded multiple bands after electrophoresis on 1% agarose gels, they were gel extracted using QIAquick Gel Extraction kit. If samples extracted from insects containing a large volume of blood did not result in bands after PCR, the extract was diluted 1 in 10 and PCR was conducted again to ensure that a PCR-inhibitor from the blood meal was not present (e.g., approach successful with sample UCR_ENT 00123869). We used primers for the 18S region to determine presence of trypanosomes in our extracts (Table 1; *Hwang et al., 2010*). PCR conditions consisted of an initial denaturation step of 94 °C for 5 min, denaturation at 94 °C for 30 s, the annealing temperature listed in Table 1 for 30 s, extension at 72 °C for

30 s repeated for 35 cycles with a final extension time at 72 °C for 10 min. For each set of PCRs conducted, we also included a blank PCR control that did not result in amplification.

## Purification, sequencing and analyzing

All PCR products were cleaned using SureClean (Bioline, London, UK) before sequencing using the Macrogen EZ-Seq service. Once sequences were obtained, the program Sequencher was used to process chromatographs. A nucleotide similarity BLAST search was then used to compare sequences to the GenBank database. Sequences of the 12S gene were considered to be derived from the same species as the closest match represented in GenBank only if they were 100% identical and most or all other members of that genus were also represented and differed in sequence (e.g., sequences KX779919, 100% to *Choloepus didactylus* (Linnaeus), and KX779929, 100% to *Neotoma lepida* Thomas) otherwise we classified it only to genus (e.g., sequence KX779923, 100% to *Lagothrix lagotricha* (Humboldt)) or to an even higher level (e.g., sequence KX779938, 100% to *Mustela kathiah* Hodgson and classified as Mustelidae sp.). When multiple members of a genus were represented in GenBank, but none matched 100%, we classified our sequence to that genus if it was more than 98% identical to one member of the genus and closer to other members of that genus than to any other genus (most sequences, e.g., sequences KX779920 98.6% to *Dasyprocta leporina* [Linnaeus] and KX779934 98.7% to *Saguinus oedipus* (Linnaeus)). Trypanosome-derived PCR products of the 18S rRNA gene were sequenced and the sequence compared to the GenBank database for identification.

## Phylogeny construction

We gathered all available data for Triatominae species and closely related reduviids (Stenopodainae, *Zelurus* spp. *Opisthacidius* spp.) on GenBank totaling 9,343 bp from 122 taxa as aligned using the MAFFT EINS-i algorithm (*Katoh & Standley, 2013*) comprising the loci from the nuclear rRNA operon and the mitochondrial genome (18S rRNA, ITS 1 5.8 rRNA and ITS 2, D2, D3-D5 regions of the 28S rRNA; 16S rRNA COI, COII, Cytb) and constructed a phylogeny using a partitioned RaxML analysis (Fig. S1, sampled species in red) partitioned by the best scheme as determined by PartitionFinder (Phylip, Data S1; nine partitions, Data S2). We excluded *Belminus herreri* Lent & Wygodzinsky from our final analysis due to the reconstruction of this taxon as sister to *T. rubrofasciata* with low support and a long terminal branch; this result may be due to this taxon only being represented by the 18S gene that has low phylogenetic utility for relatively recent divergences and the lack of that gene in the only other representative of the tribe Bolboderini in our dataset, *Microtriatoma trinidadensis* (Lent) (rRNA 28S D2 region only). We also excluded two mitochondrial loci: the cytochrome b sequence from *Panstrongylus rufotuberculatus* (Champion) that has been hypothesized to be introgressed from *Panstongylus chinai* (Del Ponte) (*Sempertegui-Sosa, 2012*) as well as a purported cytochrome oxidase II from *T. carrioni* Larrouse that shares a high similarity to members of the *T. infestans* clade. To visualize all species of Triatominae in a single figure, we placed all taxa for which reliable molecular data are currently unavailable to this backbone phylogeny using information on morphological similarities from (*Lent & Wygodzinsky, 1979*).

## Literature review

We used a Web of Science search (all databases) to query the species name of all the 152 currently recognized extant Triatominae species and surveyed results for literature records of host associations. Briefly, we assessed titles and abstracts for the inclusion of information on host associations for all matches and if we determined that they may include relevant information, we scrutinized the publication for the record of the host association and the type of method with which it was achieved. For a few widely studied triatomine species with an extremely high number of matching publications (e.g., *Triatoma infestans*, *Rhodnius prolixus*), we limited our searches with additional target words such as ''gut contents'', ''blood meal'' or ''host feed*'' to increase the feasibility of surveying all relevant results. Every attempt was made to find the primary source of a host record, but occasionally we resorted to citing reviews for which the source of the record was unclear (and may have been first reported by the authors of that review). We limited the inclusion of host records to evolutionarily relevant host species by excluding laboratory results or host associations of exclusively domestic animals such as dogs, chickens or other farmed animals due to the unnatural and recent modern presence of these hosts. An exception was made for domestic or peridomestic rodents, including mice, rats and guinea pigs that may also occur in natural environments. A detailed protocol for our literature is included as Article S1. Host taxa were divided into major groups with all arthropods, amphibians, birds and reptiles each comprising a single group and the remaining mammals split primarily by order with some exceptions (suborder level for xenarthrans, superfamily level for primates and rodents, family level for carnivores). A record of *Mepraia parapatrica* Frías-Lasserre feeding on a sea lion (*Otaria flavescens* Péron) was not visualized on the figure. While our literature review was as exhaustive as reasonably possible, it is possible that some relevant studies have not been recorded, particularly references in the grey literature (e.g., otherwise unpublished data from theses or technical reports) and others not referenced by Web of Science.

## RESULTS AND DISCUSSION

Of 64 total Triatominae specimens tested (38 males and 26 females), we were able to determine a host association for 24 specimens (37.5%) with a maximum of a single host determined per specimen. Of hosts with observable blood (28 out of 56 with observations recorded; quantity of blood categories 2–4), 18 or 64.3% gave positive host results, compared to 2 or 7.1% of those observed without visible blood (category 1) in the digestive tract for which we obtained a host sequence. Specimens with observable blood but without amplifiable host DNA may represent samples with DNA degraded beyond allowing for amplification with primers targeting the ∼160 bp region of the 12S gene. Alternatively, it is possible that the Kitano 12S primers do not amplify 12S sequences from certain hosts, though they were designed based on sequences from divergent vertebrates from sharks to fish, reptiles, amphibians and mammals (*Kitano et al., 2007*). Of the 24 specimens with host determinations, 17 or 70.8% were male, a heavily skewed ratio. Similarly, of the 28 specimens with visible blood in the digestive tract, 21 or 75% were male. We speculate that blood-fed males may be more predisposed to dispersing in flight while searching for mates
and thus be more susceptible to light traps. In contrast, females may be stationary after feeding, attempting to find a suitable place for oviposition, possibly near or in the same location as the source of a blood meal. The infection rate of all specimens with *T. cruzi* across sampled Triatominae was 31.3% or 20 specimens with one specimen of *Rhodnius pictipes* Stål producing a band that, after sequenced, matched that of *Trypanosoma rangeli* Tejera. Of the 20 specimens testing positive for *T. cruzi*, 50% also possessed detectable host DNA and 50% did not. *Panstrongylus geniculatus* was the species with the highest number of *T. cruzi* positive individuals (5/11) followed by *Triatoma protracta* (4/17) and all three individuals tested of *T. dimidiata* (Latrielle) which was the species with the highest positive percentage rate (100%; 3/3) along with the single positive specimens of *T. dispar* Lent and *T. recurva* Stål. Specimens of seven additional species also tested positive for *T. cruzi* (Table 2), all of which were previously known to be capable of hosting the parasite (*Galvão & Justi, 2015*).

We did not see evidence of double peaks in our chromatographs and direct sequencing always resulted in a single, uncontroversial host sequence, similar to results from other studies conducted using similar primers (*Gottdenker et al., 2012*), but in contrast to other studies where cloning of the PCR product was performed and as many as four hosts were detected from a single specimen (*Waleckx et al., 2014*). The archival nature of our specimens may have contributed towards the lack of detection of blood meals other than the one with most abundant DNA available. The 24 hosts detected represented mainly animals present in a sylvatic environment with the exception of three records of the domestic dog from *T. protracta* ($n = 17$) specimens collected in Southern California in residential areas, one record of a deer mouse detected from *T. pallidipennis* Stål ($n = 1$) at a residence in Mexico and two records of humans from sylvatic areas in Ecuador and Bolivia from *Rhodnius pictipes* ($n = 4$) and *Panstrongylus rufotuberculatus* ($n = 3$), respectively. These results represent fewer human records than some similar DNA-based studies (48.8% in *Waleckx et al. (2014)*; 38% in *Stevens et al. (2012)*), which may reflect the sylvatic nature of most of our collecting sites and, potentially, the rigorous sterilization protocol we utilized. Because we did not obtain any sequences from marsupials or armadillos, it is possible that the Kitano 12S primers do not amplify 12S sequences from these hosts. While the primers match the corresponding sequence from armadillos at all but two mismatches at beginning of the reverse primer, the sequence in opossums appears to have several mismatches with the reverse primer that may have prevented binding and amplification. Of the sylvatic hosts detected, many are arboreal mammals and a surprising number reflect new host species, or new records for the larger groups of vertebrates they belong to (Fig. 1, dark red rectangles) as defined in Fig. 1. Even after an extensive literature review, we were unable to find any record of these new groups of animals being recorded for hosts of these species. The new hosts detected for *Panstrongylus geniculatus* ($n = 11$) include sloths (*Choloepus didactylus* Linnaeus), the arboreal weasel-like animal known as a tayra (*Eira barbara* (Linnaeus)), an unidentified member of Mustelidae, New World monkeys (*Lagothrix* sp. and *Saguinus* sp.) and agoutis (*Dasyprocta* sp.), all but the last, representing the first record for that group of vertebrate for that species. The tayra record represents the first record of any Triatominae species feeding on this species though it has been previously

**Table 2** Specimen data including data obtained by vertebrate and trypanosome DNA targeted PCR.

| USI | Species | COUNTRY: primary subdivision | Ecotype | Collecting date (year) | EtOH concentration | Sex | Quantity of blood (1–4) | Host 12S sequence | Top BLAST hit of host 12S sequence | Inferred host | Trypanosome sequence | Top trypanosome BLAST hit |
|---|---|---|---|---|---|---|---|---|---|---|---|---|
| UCR_ENT 00012958 | *Eratyrus mucronatus* | PERU; Loreto | Sylvatic | 2007 | ? | M | NA | | | | | |
| UCR_ENT 00012937 | *Opisthacidius parkoi* | PERU; Loreto | Sylvatic | 2007 | ? | F | NA | | | | | |
| UCR_ENT 00119043 | *Panstrongylus geniculatus* | FRENCH GUIANA; Cayenne | Sylvatic | 2010 | 95% | M | 4 | KX779934 | 98.7% to *Saguinus oedipus* | *Saguinus* sp. (Tamarin) | | |
| UCR_ENT 00119044 | *Panstrongylus geniculatus* | FRENCH GUIANA; Cayenne | Sylvatic | 2010 | 95% | M | 3 | KX779919 | 100% to *Choloepus didactylus* | *Choloepus didactylus* (Southern two-toed sloth) | | |
| UCR_ENT 00119045 | *Panstrongylus geniculatus* | FRENCH GUIANA; Cayenne | Sylvatic | 2010 | 95% | M | 1 | | | | | |
| UCR_ENT 00119046 | *Panstrongylus geniculatus* | FRENCH GUIANA; Cayenne | Sylvatic | 2010 | 95% | F | 1 | KX779920 | 98.6% to *Dasyprocta leporina* | *Dasyprocta* sp. (Agouti) | KX779898 | 100% to *Trypanosoma cruzi* |
| UCR_ENT 00119047 | *Panstrongylus geniculatus* | FRENCH GUIANA; Cayenne | Sylvatic | 2010 | 95% | F | 1 | | | | | |
| UCR_ENT 00012932 | *Panstrongylus geniculatus* | PERU; Loreto | Sylvatic | 2007 | ? | M | 4 | KX779923 | 100% to *Lagothrix lagotricha* | *Lagothrix* sp. (Wooly monkey) | KX779901 | 100% to *Trypanosoma cruzi* |
| UCR_ENT 00012959 | *Panstrongylus geniculatus* | PERU; Loreto | Sylvatic | 2007 | ? | M | 3 | KX779927 | 99.3% to *Eira barbara* | *Eira barbara* (Tayra) | | |
| UCR_ENT 00063360 | *Panstrongylus geniculatus* | NICARAGUA; Rio San Juan | Sylvatic | 2010 | 95% | M | 3 | KX779922 | 97.1% to *Dasyprocta punctata* | *Dasyprocta* sp. (Agouti) | KX779905 | 100% to *Trypanosoma cruzi* |
| UCR_ENT 00063367 | *Panstrongylus geniculatus* | BOLIVIA; Santa Cruz | Sylvatic | 2009 | 95% | M | 2 | | | | | |
| UCR_ENT 00119054 | *Panstrongylus geniculatus* | COSTA RICA; Heredia | Sylvatic | 2013 | 95% | M | 2 | KX779938 | 100% to *Mustela kathiah* | Mustelidae sp. (Weasels, badgers, otters and allies) | KX779914 | 100% to *Trypanosoma cruzi* |
| UCR_ENT 00123859 | *Panstrongylus geniculatus* | PANAMA; Colon | Sylvatic | 2008 | ? | F | 1 | | | | KX779915 | 100% to *Trypanosoma cruzi* |
| AMNH_PBI 00021872 | *Panstrongylus lignarius* | FRENCH GUIANA; Montsinery | Sylvatic | 2004 | ? | M | 2 | | | | KX779910 | 100% to *Trypanosoma cruzi* |
| UCR_ENT 00012933 | *Panstrongylus rufotuberculatus* | PERU; Loreto | Sylvatic | 2007 | ? | M | 3 | KX779933 | 99.4% to *Ateles belzebuth* | *Ateles* sp. (Spider monkey) | | |

Georgieva et al. (2017), *PeerJ*, DOI 10.7717/peerj.3826

**Table 2** (*continued*)

| USI | Species | COUNTRY: primary subdivision | Ecotype | Collecting date (year) | EtOH concentration | Sex | Quantity of blood (1–4) | Host 12S sequence | Top BLAST hit of host 12S sequence | Inferred host | Trypanosome sequence | Top trypanosome BLAST hit |
|---|---|---|---|---|---|---|---|---|---|---|---|---|
| UCR_ENT 00063350 | *Panstrongylus rufotuberculatus* | BOLIVIA; Santa Cruz | Sylvatic | 2009 | 95% | M | 1 | KX779925 | 100% to *Homo sapiens* | *Homo sapiens* (Human) | | |
| UCR_ENT 00012961 | *Panstrongylus rufotuberculatus* | PERU; Loreto | Sylvatic | 2007 | ? | M | 2 | | | | | |
| UCR_ENT 00063364 | *Panstrongylus rufotuberculatus* | NICARAGUA; Rio San Juan | Sylvatic | 2010 | 95% | M | 3 | KX779940 | 98.5% to *Coendou prehensilis* | *Coendou* sp. (Prehensile-tailed porcupine) | KX779907 | 100% to *Trypanosoma cruzi* |
| UCR_ENT 00123860 | *Paratriatoma hirsuta* | MEXICO; Baja California Norte | Sylvatic | 2009 | 95% | M | 2 | KX779929 | 100% to *Neotoma lepida* | *Neotoma lepida* (Desert woodrat) | | |
| UCR_ENT 00003254 | *Paratriatoma hirsuta* | USA; California | Peridomestic | 2009 | 95% | M | 4 | | | | | |
| UCR_ENT 00012924 | *Rhodnius barretti* | PERU; Loreto | Sylvatic | 2007 | ? | F | 1 | KX779937 | 99.3% to *Saimiri sciureus* | *Saimiri* sp. 3 (Squirrel monkey) | | |
| UCR_ENT 00012925 | *Rhodnius barretti* | PERU; Loreto | Sylvatic | 2007 | ? | M | 1 | | | | | |
| UCR_ENT 00012929 | *Rhodnius barretti* | PERU; Loreto | Sylvatic | 2007 | ? | M | 4 | KX779918 | 100% to *Saimiri sciureus* | *Saimiri* sp. 1 (Squirrel monkey) | | |
| UCR_ENT 00012928 | *Rhodnius barretti* | PERU; Loreto | Sylvatic | 2007 | ? | F | 1 | | | | | |
| UCR_ENT 00002734 | *Rhodnius barretti* | ECUADOR; Orellana | Sylvatic | 2009 | 95% | M | 3 | KX779921 | 99.4% to *Saimiri sciureus* | *Saimiri* sp. 2 (Squirrel monkey) | KX779912 | 100% to *Trypanosoma cruzi* |
| UCR_ENT 00002735 | *Rhodnius barretti* | ECUADOR; Orellana | Sylvatic | 2009 | 95% | F | 1 | | | | KX779900 | 100% to *Trypanosoma cruzi* |
| UCR_ENT 00119053 | *Rhodnius barretti* | PERU; Madre de Dios | Sylvatic | 2005 | 95% | F | 4 | KX779924 | 99.4% to various *Cebus* spp. | *Cebus* sp. 2 (Gracile capuchin monkey) | | |
| UCR_ENT 00063365 | *Rhodnius pallescens* | NICARAGUA; Rio San Juan | Sylvatic | 2010 | 95% | M | 1 | | | | | |
| UCR_ENT 00002736 | *Rhodnius pictipes* | ECUADOR; Orellana | Sylvatic | 2009 | 95% | F | 1 | KX779936 | 100% to *Homo sapiens* | *Homo sapiens* (Human) | | |
| UCR_ENT 00119039 | *Rhodnius pictipes* | FRENCH GUIANA; Cayenne | Sylvatic | 2010 | 95% | F | 1 | | | | | |

Peerj

| USI | Species | COUNTRY: primary subdivision | Ecotype | Collecting date (year) | EtOH concentration | Sex | Quantity of blood (1–4) | Host 12S sequence | Top BLAST hit of host 12S sequence | Inferred host | Trypanosome sequence | Top trypanosome BLAST hit |
|-----|---------|------------------------------|---------|------------------------|--------------------|-----|-------------------------|-------------------|-----------------------------------|---------------|---------------------|----------------------------|
| UCR_ENT 00119040 | Rhodnius pictipes | FRENCH GUIANA; Cayenne | Sylvatic | 2010 | 95% | F | 1 | | | | | |
| UCR_ENT 00119041 | Rhodnius pictipes | FRENCH GUIANA; Cayenne | Sylvatic | 2010 | 95% | M | 4 | | | | KX779911 | 100% to Trypanosoma rangeli |
| UCR_ENT 00063354 | Rhodnius robustus | BOLIVIA; Santa Cruz | Sylvatic | 2009 | 95% | F | 4 | KX779932 | 99.4% to Nasua nasua | Nasua sp. (Coati) | KX779904 | 100% to Trypanosoma cruzi |
| UCR_ENT 00123871 | Triatoma dimidiata | COSTA RICA; Heredia | Sylvatic | 2010 | 95% | M | 4 | | | | KX779899 | 100% to Trypanosoma cruzi |
| UCR_ENT 00123870 | Triatoma dimidiata | COSTA RICA; Heredia | Sylvatic | 2010 | 95% | M | 1 | | | | KX779897 | 100% to Trypanosoma cruzi |
| UCR_ENT 00063362 | Triatoma dimidiata | NICARAGUA; Rio San Juan | Sylvatic | 2010 | 95% | M | 4 | KX779917 | 100% to various Cebus spp. | Cebus sp. 1 (Gracile capuchin monkey) | KX779913 | 100% to Trypanosoma cruzi |
| UCR_ENT 00052210 | Triatoma dispar | COSTA RICA; Alajuela | Sylvatic | 2008 | 95% | M | 2 | KX779931 | 98.2% to Potos flavus | Procyonidae sp. (Raccoons, kinkajous, coatis, olingos and allies) | KX779902 | 100% to Trypanosoma cruzi |
| UCR_ENT 00119051 | Triatoma lecticularia | USA; Texas | Sylvatic | 2009 | 95% | F | 3 | KX779935 | 100% to Cathartes aura | Cathartes aura (Turkey vulture) | KX779908 | 100% to Trypanosoma cruzi |
| UCR_ENT 00119052 | Triatoma lecticularia | USA; Texas | Sylvatic | 2009 | 95% | M | 1 | | | | | |
| UCR_ENT 00123526 | Triatoma mexicana | MEXICO: Guerrero | Domestic | 2011 | ? | F | 4 | | | | | |
| UCR_ENT 00119055 | Triatoma mexicana | MEXICO: Guerrero | Domestic | 2011 | ? | F | 4 | | | | | |
| UCR_ENT 00123869 | Triatoma pallidipennis | MEXICO: Guerrero | Domestic | 2011 | ? | M | 4 | KX779926 | 99.4% to Peromyscus maniculatus | Peromyscus sp. (Deer mouse) | | |
| UCR_ENT 00005163 | Triatoma protracta | USA; California | Peridomestic | 2008 | 95% | F | NA | | | | KX779909 | 100% to Trypanosoma cruzi |
| UCR_ENT 00005165 | Triatoma protracta | USA; California | Peridomestic | 2008 | 95% | M | NA | KX779930 | 100% to Canis lupus | Canis lupus (Dog) | | |
| UCR_ENT 00005167 | Triatoma protracta | USA; California | Peridomestic | 2008 | 95% | F | NA | | | | KX779906 | 100% to Trypanosoma cruzi |
| UCR_ENT 00005184 | Triatoma protracta | USA; California | Peridomestic | 2008 | 95% | M | NA | | | | | |

**Table 2** (*continued*)

| USI | Species | COUNTRY: primary subdivision | Ecotype | Collecting date (year) | EtOH concentration | Sex | Quantity of blood (1–4) | Host 12S sequence | Top BLAST hit of host 12S sequence | Inferred host | Trypanosome sequence | Top trypanosome BLAST hit |
|---|---|---|---|---|---|---|---|---|---|---|---|---|
| UCR_ENT 00005185 | *Triatoma protracta* | USA; California | Peridomestic | 2008 | 95% | F | NA | | | | KX779896 | 100% to *Trypanosoma cruzi* |
| UCR_ENT 00005186 | *Triatoma protracta* | USA; California | Peridomestic | 2008 | 95% | M | NA | | | | KX779903 | 100% to *Trypanosoma cruzi* |
| UCR_ENT 00003031 | *Triatoma protracta* | USA; California | Peridomestic | 2009 | 95% | F | 1 | | | | | |
| AMNH_PBI 00218748 | *Triatoma protracta* | USA; California | Peridomestic | 2007 | 95% | F | 1 | | | | | |
| AMNH_PBI 00218756 | *Triatoma protracta* | USA; California | Peridomestic | 2007 | 95% | M | 1 | | | | | |
| AMNH_PBI 00218755 | *Triatoma protracta* | USA; California | Peridomestic | 2007 | 95% | F | NA | KX779928 | 100% to *Canis lupus* | *Canis lupus* (Dog) | | |
| UCR_ENT 00123868 | *Triatoma protracta* | USA; California | Peridomestic | 2014 | 95% | M | 2 | | | | | |
| UCR_ENT 00123866 | *Triatoma protracta* | USA; California | Peridomestic | 2008 | 95% | M | 1 | | | | | |
| UCR_ENT 00123867 | *Triatoma protracta* | USA; California | Peridomestic | 2008 | 95% | M | 3 | KX779939 | 100% to *Canis lupus* | *Canis lupus* (Dog) | | |
| UCR_ENT 00123865 | *Triatoma protracta* | USA; California | Peridomestic | 2008 | 95% | M | 2 | | | | | |
| UCR_ENT 00123864 | *Triatoma protracta* | USA; California | Peridomestic | 2008 | 95% | F | 1 | | | | | |
| UCR_ENT 00123863 | *Triatoma protracta* | USA; California | Peridomestic | 2008 | 95% | M | 1 | | | | | |
| UCR_ENT 00123862 | *Triatoma protracta* | USA; California | Peridomestic | 2008 | 95% | F | 1 | | | | | |
| UCR_ENT 00119056 | *Triatoma recurva* | USA; Arizona | Sylvatic | 2014 | 95% | F | 1 | | | | KX779916 | 100% to *Trypanosoma cruzi* |
| UCR_ENT 00119038 | *Triatoma rubida* | USA; Arizona | Sylvatic | 2008 | ? | F | 1 | | | | | |
| UCR_ENT 00119048 | *Triatoma rubida* | USA; Arizona | Sylvatic | 2006 | ? | F | 1 | | | | | |
| UCR_ENT 00123861 | *Triatoma ryckmani* | GUATEMALA; Peten | Sylvatic | 2015 | 95% | F | 1 | | | | | |
| UCR_ENT 00119049 | *Triatoma sanguisuga* | USA; Florida | Sylvatic | 2011 | 95% | M | 1 | | | | | |
| UCR_ENT 00119050 | *Triatoma sanguisuga* | USA; Florida | Sylvatic | 2011 | 95% | M | 1 | | | | | |

known to be infected with *T. cruzi* (*Ferreira & Deane, 1938*). We recovered several new host records for New World monkeys, apart from the two species identified for *P. geniculatus,* including *Saimiri* sp., *Ateles* sp., and *Cebus* sp. For *Rhodnius barretti* ($n = 7$), *Panstrongylus rufotuberculatus* ($n = 3$) and *T. dimidiata* ($n = 3$), respectively. These all represent the first New World monkey hosts recorded for those species except for *T. dimidiata* which has been previously noted as possessing antigens reacting to antibodies developed for detecting New World monkey specific proteins. Our new host records also included the arboreal porcupine genus *Coendou* for *P. rufotuberculatus*, coati (*Nasua* sp.) for *R. robustus* ($n = 1$) and a sequence with the closest match to a kinkajou (*Potos sp.*) but only determined to the level of Procyonidae for *Triatoma dispar* ($n = 1$), all new animal groups for those species. The prevalence of arboreal mammals among our new host records may reflect the inaccessibility of these habitats for gathering associational observations or absence of antigens targeting these taxa in previous studies. For example, while various populations of non-human primates are known to have high infection rates with *T. cruzi* (*Lisboa et al., 2015*) and were even one of the first mammals found to be infected with this parasite (*Deane, 1964*), only six species of Triatominae had previously been recorded in association with New World monkeys with four new records (and three new species) in this study alone. Potentially, these under recognized groups of hosts play a more prominent role in the sylvatic cycle of *T. cruzi* than has been previously understood.

As a result of our literature review, we were able to detect some feeding patterns that have not been previously widely recognized. However, it should be noted that host preferences cannot be determined with this kind of data nor does the lack of evidence for a particular host suggest that this host association may not be uncovered in the future. All of our aggregated data is reported in Table S1 with full references in Article S2. The number of species with associations with amphibians (14), reptiles (35) and birds (55, or 36% of all Triatominae species) is higher than we expected as these species tend to be thought of as minor hosts for Triatominae or are primarily associated only with certain species (*Rabinovich et al., 2011*). This discrepancy may reflect that these hosts are not known to be able to harbor the *T. cruzi* pathogen (*Kierszenbaum, Gottlieb & Budzko, 1981*; *Urdaneta-Morales & McLure, 1981*) and deliberate attention has been paid to mammalian reservoir hosts. The most common group of known hosts among all species of Triatominae are marsupials (55 species), birds (55 species), humans (51) and Muroidea including rats and mice (47 species). Some of the rarest groups of known sylvatic hosts are shrews, artiodactyls, felines and sea lions, each known only from a single host record, although this excludes records for domestic species classified in these groups such as the domestic cat and the domestic pig. There are 40 kissing bug species for which we were unable to find any records for sylvatic hosts. The largest group of Triatominae with scarce host data are the Old World species including the genus *Linshcosteus* and the *T. rubrofasciata* species group. Species which have been studied using DNA-based methods tend to have a broader range of known hosts than species only investigated using associational and immunological approaches. For example, *Rhodnius pallescens* Barber and *T. gerstaeckeri* have the broadest ranges of recorded host groups (20 and 15 respectively, out of 26 groups recorded across all species) primarily because of single studies focused on each of those species
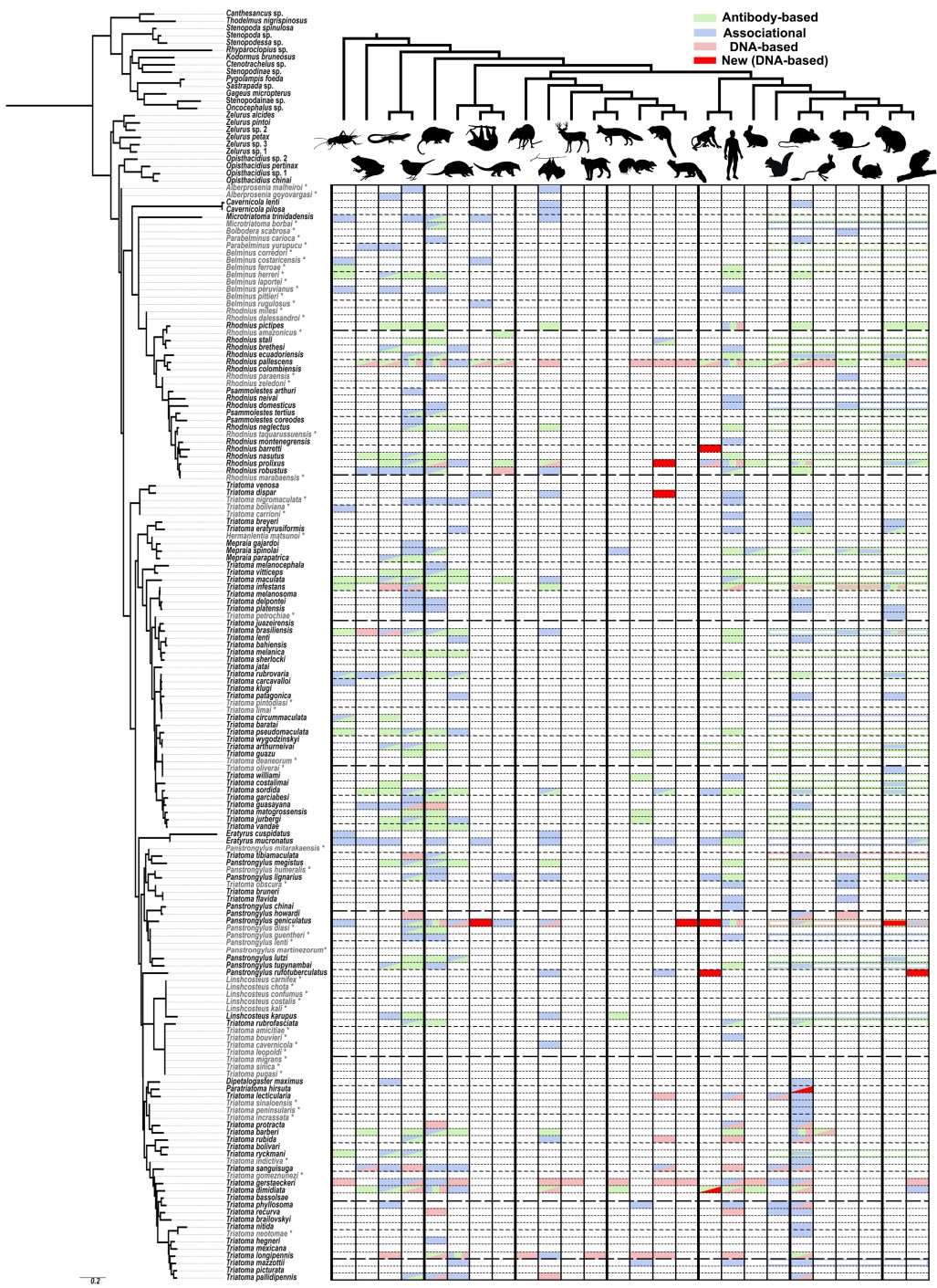

**Figure 1   A visualization of the known sylvatic hosts of Triatominae and the type of record(s) support-ing that association.** The animals at the top of the figure represent the following groups from left to right, alternating from top to bottom: arthropod (Arthropoda), amphibian (Amphibia), lizards (Lepidosauro-morpha), bird (Aves), opossums (Didelphimorphia), armadillo (Cingulata), sloth (Folivora), anteater (Vermilingua), shrew (Eulipotyphla), bat (Chiroptera), even-toed ungulate (Artiodactyls), feline (Felidae), canine (Canidae), musteloid (Mephitidae; Skunk), (continued on next page…)

**Figure 1 (…continued)**
musteloid (Procyonidae; Racoons and relatives), musteloid (Mustelidae; Weasel), platyrrhine monkey (Ceboidea), human (Hominidae), rabbits (Lagomorpha), rodent (Sciuromorpha), rodent (Muroidea), rodent (Geomyoidea), rodent (Octodontoidea), rodent (Chinchilloidea), rodent (Cavioidea), rodent (Erethizontoidea). Relationships among mammals were simplified from *Nyakatura & Bininda-Emonds (2012)* for Carnivora, *Foley, Springer & Teeling (2016)* for deep level relationships and *Fabre et al. (2012)* for rodents. Triatominae taxa in light grey and indicated with an asterisk were added to the phylogeny based on morphological similarities indicated in the literature. Filled in matrix rectangle indicate observations with green indicating antibody-based observations; blue, associational; and red, DNA-based with bright red indicating newly obtained DNA-based host records for that species in our study. Colored outlines of rectangles across all represented rodent superfamilies represent observations specified only to a monophyletic higher level, e.g., the order Rodentia or Primates.

(*Gottdenker et al., 2012*; *Gorchakov et al., 2016*). Some species thought to have narrow host ranges show some evidence, although mostly associational and rarely antibody-based, of also feeding on additional hosts such as rodents for *Cavernicola* (thought to associate with bats) and both rodents and marsupials for *Psammolestes* (thought to associate with birds). There is no obvious host specificity of Triatominae clades and, overall, our review points towards the generalist bent of the great majority Triatominae species, perhaps influenced more by hosts present in a given habitat and microhabitat than innate preference towards certain groups of hosts.

A comprehensive understanding of the ecology of *Trypanosoma cruzi* and relevant vectors, including primarily sylvatic Triatominae species, will be important in further controlling Chagas disease. One important consideration in any eventual attempt to eradicate Chagas disease is that oral transmission in the sylvatic cycle may be the primary mechanism of parasite spread (*Roque et al., 2008*) and some insectivorous mammals often have a higher prevalence of the disease (*Hodo & Hamer, 2017*). It has been reported that even *T. cruzi*-infected vertebrates eaten as prey can be infective (*Thomas, Rasweiler & D'Alessandro, 2007*) and it appears that Chagas disease can "bioaccumulate" in predators including highly carnivorous top predators (e.g., ocelots), that are nocturnal without permanent dens and are thus unlikely to be exposed to feeding Triatominae (*Rocha et al., 2013*). Predators that probably are frequently exposed to Triatominae, for example nest-building arboreal species like coatis and tayras, play an important role in the maintenance and spread of *T. cruzi*. For that reason, it has been predicted that if there was development of a successful vaccine against *T. cruzi* in the future, targeting such predators for vaccines among other highly-infected host species could theoretically be an effective strategy in limiting the disease (*Stella et al., 2017*). Thus, the confirmation of Triatominae feeding on certain carnivores near the top of the food web can be of major importance in understanding and limiting the sylvatic cycle of *T. cruzi*.

New approaches for modeling networks of vector host interactions based on the co-occurrence of mammal and Triatominae species offer a promising method for understanding Triatominae ecology (*Ibarra-Cerdeña et al., 2017*; *Rengifo-Correa et al., 2017*). Combining such studies with reviews of *T. cruzi*-infected mammals (*Browne et al., 2017*) as well as documented host associations of individual species such as provided in our review offers a route for advancing our understanding of the spread of *Trypanosoma*

*cruzi.* While each individual study does not manage to sample a completely representative population of vectors and is limited in their recovery of host identity by methodology-specific biases, a synthetic record of host associations overcomes many shortcomings.

## CONCLUSION

We believe that DNA-based methods for determining host associations of blood-feeding species offer the best route forward in understanding the biology and epidemiological importance of this speciose group of vectors. While there may be bias in representation of known vertebrates on public databases, these databases continue to encompass data for an ever-growing range of species. We recommend the deposition of sequences acquired from vertebrate hosts of vectors into public databases even if their identity is not known at the time, a practice not yet routinely conducted, as the influx of DNA data may later shed light on their identity. While DNA-based studies of blood meals can suffer from primer specificity biases, rigorous testing of primers sets for universality and specificity can minimize this possibility. We have shown that this method can be used on archival specimens and we recommend future efforts using this relatively cheap, efficient and effective method in order to better understand the habits of all species of these vectors, particularly those that are less well studied or rarely collected. PCR on archival specimens may sometimes give false negative results or only allow sequencing of the most recent host, but this is preferable to over reactivity of antibody based tests (*Barrett, 1991*) and can be overcome given a sufficiently large enough sample size. We found that 95% ethanol seemed to preserve DNA well enough for amplification of the 150 bp chunk of 12S rRNA using our preferred set of primers (*Kitano et al., 2007*). While we did achieve some results for specimens without observable blood in the digestive tract, we had a much higher success rate for those with observable blood and time and energy could be saved by not extracting DNA from the former group of specimens. We hope that the application of this approach in combination with extensive sampling of additional Triatominae, especially in sylvatic environments, can offer critical and robust data on potential vectors and help to impede the spread of Chagas disease.

## ACKNOWLEDGEMENTS

We thank Jason Cryan, Andrew Ernst, Dimitri Forero, Sarah Frankenberg, Jeremy Huff, Warren Macdonald, Michael E. Irwin, JJ Ramirez, Gavin Svenson, Julie Urban, and Michael Whiting for donations of specimens; and two reviewers for valuable suggestions that greatly improved this manuscript.

### Funding

This study was supported by a University of California, Riverside Academic Senate Grant to Christiane Weirauch. The funders had no role in study design, data collection and analysis, decision to publish, or preparation of the manuscript.

## Grant Disclosures

The following grant information was disclosed by the authors:
University of California, Riverside Academic Senate Grant to Christiane Weirauch.

## Competing Interests

The authors declare there are no competing interests.

## Author Contributions

- Anna Y. Georgieva and Eric R.L. Gordon conceived and designed the experiments, performed the experiments, analyzed the data, wrote the paper, prepared figures and/or tables, reviewed drafts of the paper.
- Christiane Weirauch conceived and designed the experiments, contributed reagents/materials/analysis tools, wrote the paper, reviewed drafts of the paper.

## Data Availability

Newly acquired sequences are available with GenBank accession numbers KX779896–KX779940.

## Supplemental Information

Supplemental information for this article can be found online at http://dx.doi.org/10.7717/peerj.3826#supplemental-information.

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
