# Peer review of "Sylvatic host associations of Triatominae and implications for Chagas disease reservoirs: a review and new host records based on archival specimens"

_PeerJ, doi:10.7717/peerj.3826_

## Round 0.1 · original submission · Major Revisions

· Academic Editor

Major Revisions

Both colleagues that reviewed your manuscript have much appreciated your work, as well as myself. They underline, however, a series of points that need your attention, in particular in relation with the bibliographic basis of your work.

[# Staff Note: It is PeerJ policy that references should only be added if the authors agree with the reviewers that they are relevant for the article #]

Reviewer 1 ·

Basic reporting

Please see 'General comments' below

Experimental design

Please see 'General comments' below

Validity of the findings

Please see 'General comments' below

Comments for the author

In this manuscript, Georgieva, Gordon and Weirauch present an assessment of putative associations between triatomine bugs, the vectors of Trypanosoma cruzi, and (mainly wild) vertebrate hosts in the Americas. The authors combine a review of the literature with molecular bloodmeal identification in a sample of archival triatomines. The report is within the scope of the journal and is overall well written. I think, however, that parts of it need improvement.
My main concern about the manuscript is that the obvious key limitations of this and similar studies should be more clearly stated and more thoroughly discussed. Although the authors do mention some important caveats, I think that readers will need further explanation to be able to gauge the strengths and weaknesses of this report. The key limitations that I think need more elaboration include, at least, what I’d call “multilevel bias”, which may be described as follows.
First, the review of the literature on putative triatomine-vertebrate host associations seems to have missed some studies. I did not go through that literature in any systematic way, but I can provide a few examples. For bloodmeal analyses, Mascarenhas 1991 Bol Museu Paraense Emilio Goeldi ser Zool 7:107-16 (Rhodnius brethesi); Abad-Franch et al. 2002 Mem Inst Oswaldo Cruz 97:199-202, doi: 10.1590/S0074-02762002000200010 (R. ecuadoriensis); Pinto et al. 2008 Rev Peru Med Exp Salud Publica 25:179-84, http://www.scielo.org.pe/pdf/rins/v25n2/a04v25n2.pdf (Panstrongylus herreri); Farfan-Garcia & Angulo-Silva 2011 Rev Salud Publica Bogota 13:163-72, doi: 10.1590/S0124-00642011000100014 (Triatoma dimidiata); Peña-Garcia et al. 2014 Am J Trop Med Hyg 91:1116-24, doi: 10.4269/ajtmh.14-0112 (mainly R. prolixus); Villacis et al. 2015 Am J Trop Med Hyg 92:187-92, doi: 10.4269/ajtmh.14-0250 (P. howardi; also with associational observations); or Hernández et al. 2016 Parasit Vectors 9:620, doi: 10.1186/s13071-016-1907-5 (six bug species). Examples of association studies include Botto-Mahan et al. 2005 Mem Inst Oswaldo Cruz 100:237-9, doi: S0074-02762005000300003; Botto-Mahan et al. 2005 Acta Trop 5:160-3, doi: 10.1016/j.actatropica.2005.05.001 (both Mepraia spinolai; see also Chacon Olivos 2016 at http://repositorio.uchile.cl/bitstream/handle/2250/141205/Determinacion-del-perfil-alimentario-de-triatominos-silvestres-en-zonas-endemicas-de-Chile.pdf?sequence=1); Valença-Barbosa et al. 2014 PLoS Negl Trop Dis 8:e2861, doi: 10.1371/journal.pntd.0002861; Valença-Barbosa et al. 2014 Am J Trop Med Hyg 90:1059-62, doi: 10.4269/ajtmh.13-0204 (both T. brasiliensis); or Lopez-Cancino et al. 2015 Acta Trop 151:58-72, doi: 10.1016/j.actatropica.2015.07.021 (T. dimidiata). The review by Barrett (1991 Adv Dis Vector Res 8: 143-76, doi: 10.1007/978-1-4612-3110-3_6) is also informative. For example, there one reads that Alberprosenia goyovargasi was found “…associated with lizards and snakes in galleries excavated by passalid beetles in tree trunks…” (p. 144), that Microtriatoma trinidadensis caught in “…entanglements of lianas…” had a “…pale fluid in the midgut…” and “…are often associated with detritivorous caterpillars…” (p. 146), or that Eratyrus mucronatus is found inside large living hollow trees, and particularly “…at the bases of hollow branches containing porcupine quills, rodent nests, or shelters of Potos flavus (Procyonidae). Externally, the tree forks are used as resting places by sloths and monkeys…” (p. 150).
Note also that a third “kind” of studies on triatomine-vertebrate host associations has recently emerged – one that uses data-mining techniques to try to draft and (to some extent) disentangle association networks (see Rengifo-Correa et al. 2017 Parasitology 144:760-72, doi: 10.1017/S0031182016002468 and Ibarra-Cerdeña et al. 2017 in this very journal – PeerJ 5:e3152, doi: 10.7717/peerj.3152). Finally, I’d probably also take other data sources into account, at least for the discussion; for example, Browne et al. (2017 Sci Data 4:170050, doi: 10.1038/sdata.2017.50) have recently compiled an impressive database on T. cruzi infection in vectors and vertebrates (see also http://dx.doi.org/10.5061/dryad.93mn0). In view of the emphasis given by the authors to the detection of Eira barbara DNA in some bugs’ samples, it should perhaps be noted that there are several reports of T. cruzi-infected tayras since Ferreira and Deane first published on this in 1938 (Brazil Medico 52:1159-61; see also, e.g., Barretto and Ribeiro 1972 Rev Bras Biol 32:413-8). Infection by T. rangeli can also be informative on associations between (mainly) Rhodnius and vertebrates (e.g., Miles et al. 1983 Am J Trop Med Hyg 32:1251-9, doi: 10.4269/ajtmh.1983.32.1251).
To minimize this likely literature-sampling bias I would recommend that a formal systematic review protocol be developed and applied, including explicit sub-protocols for publication searches (databases, dates, search arguments, references and citing documents of the papers retrieved, etc.), selection (inclusion and exclusion criteria and their application to database search results), data extraction (including quality assessment), and data analysis. The protocol should then be made available as supplementary material – along with lists of documents that were either reviewed or retrieved in searches but excluded from review. Note that for this specific topic it seems likely that some (perhaps many) relevant reports have been published in journals (or other media) that are not indexed by the Web of Science; inclusion of other literature databases such as PubMed, SciELO, LILACS, or Google Scholar would increase the chances of retrieving those reports including theses and other ‘gray literature’. The BibTri database (http://bibtri.com.ar/) can also be helpful.
Second, it seems inevitable that the bugs included in this and similar studies tend to comprise a biased sample of the populations they are supposed to represent. Think for example of how probable is that any sample includes wild bug populations from truly remote sites – which, incidentally, are more likely to resemble the habitats where triatomines lived (and developed their host associations, if any) before humans reached the Americas.
Third, what is really being sampled within specimens are the bugs’ recent bloodmeals, which may or may not represent true feeding preferences – if these exist at all: many triatomines, for what we know, tend to be quite opportunistic.
And, fourth, the methods used to detect and identify bloodmeal sources in each sample are all imperfect. Apart from the fact that not all possible hosts are tested, especially in antibody-based studies, this includes the (likely) possibility of false negative results (e.g. because of PCR inhibition) and false positive results (particularly for the immunological methods). And, to make things a bit worse, note that the use of adequate controls is very rarely reported in the literature on triatomine bloodmeal sources – including, by the way, this manuscript. By “adequate controls” for tests without 100% sensitivity or specificity I mean, for example, positive controls (say, from lab-reared bugs fed on known hosts), negative controls (e.g., guts sampled from unfed first-instar nymphs or perhaps from non-blood-feeding reduviid bugs), blank controls (without DNA), and internal controls (preferably same-tube, with, e.g., primers that should amplify a fragment of a triatomine gene). (These methodological shortcomings led Barrett (1991, see above; p. 164) to avoid reviewing the literature on bloodmeal identification through immunological tests, “partly because many workers found it unnecessary to mention any kind of controls or precautions against cross-contamination of samples, even when reporting unusually high rates of multiple mixed feeds or improbable host associations”. Things do not seem to have improved much since Barrett’s review was published.) Alternatively, one might think of using some of the data analysis methods developed for assessing the performance of diagnostic tests when no gold standard technique is available – perhaps latent class analysis (see for example Pepe & Janes 2007 Biostatistics 8:474-84, doi: 10.1093/biostatistics/kxl038).
Although I’m not an expert on this specific issue, I also think that not all immunological methods used for bloodmeal identification are equally reliable. Can, for example, precipitin and ELISA assays be directly compared? Specifically, are there any estimates of the sensitivity and specificity of these techniques in the context of bloodmeal identification?
These considerations about the several sources of bias that likely plague the literature on this topic make me think that, apart from “DNA-based methods” (Conclusion in the Abstract), what we really need if we want to improve our understanding of triatomine-host associations is much better field data, much better sampling designs, and better lab protocols and practices. In the meantime, the conclusions we can draw have to be clearly put in the context of the many limitations of the data.
Another point that I think should be addressed is the relationship of the manuscript to the paper by Rabinovich and colleagues (ref. 54), which appears to be cited just once (line 382) but bears important similarities to the (precipitin) review part of this report. The originality of the contribution of the present manuscript with respect to that paper must be clearly described so that readers can assess it.
I finally note that I found some formal oversights in this manuscript; perhaps the most obvious one is that some references appear more than once in the list. I spotted some other typos and mistakes; I also make some minor comments below.
l. 13 and elsewhere: it seems that some new species of Triatominae have been described recently, so that the number (“148”) is already outdated
l. 15-6: it is somewhat unclear to me how exactly “An understanding of the natural hosts of this speciose group of blood-feeding insects” might “aid ongoing efforts to impede the spread of Chagas disease”; incidence has been significantly reduced by large-scale control programs based on indoor insecticide spraying and screening of donations to blood banks – two complementary approaches for which knowledge on possible host associations does not appear to be required
l. 59: delete one “upon”
l. 61: the statement that R. robustus tends to feed on humans does not seem to be well supported by ref. 10, in which the bugs were sampled only from palm trees and no bloodmeal analyses are reported; the several R. robustus cryptic taxa are primarily silvatic and appear to be unable to establish viable breeding colonies inside houses (e.g., Fitzpatrick et al. 2008 PLoS Negl Trop Dis 2:e210, doi: 10.1371/journal.pntd.0000210).
l. 64 and 463: Romaña
Just a curiosity on non-blood food sources: recently, Diaz-Albiter et al. (2016 Parasit Vectors 9:114, doi: 10.1186/s13071-016-1401-0) have reported on sugar and plant (tomato) feeding by lab-reared Rhodnius prolixus
l. 106: gerstaeckeri
l. 245-7: please check this sentence, in which we read "...based on sequences from divergent vertebrate sequences from sharks to fish [...] (Kitano 2007)"
Table 2: “Eira 13imidia” [?]
Table 2: I see in the IUCN website that Mustela kathiah is an Asian species (http://maps.iucnredlist.org/map.html?id=41655); any comment on this?
Table 2, l. 300 and Figure: As far as we know, “Rhodnius prolixus” does not occur naturally in Bolivia; the coati bloodmeal from this specimen should most likely be assigned to R. robustus (which look very much like prolixus and do occur in eastern Bolivia)
Figure 1: why are some cell lines different from the rest (some dashed, some dotted, some colored etc.)?
Figure 1: please check whether the labels for R. montenegrensis and R. barretti (not “barreti”) in the tree are correct
Figure 1: it’s very odd that the bug labeled as “T. carrioni” (a species known from the Andes in Ecuador-Peru) clusters with “T. vandae” etc. (from Brazil-Argentina) and not with the T. venosa group (probably mislabeling or misidentification)
Figure 1: similarly, P. rufotuberculatus is unlikely to be sister to P. howardi; since, however, there seems to be some evidence that both (and P. chinai) may sporadically interbreed in nature, I guess this relationship was based on (introgressed) mitochondrial sequences (see Sempertegui-Sosa 2012 at https://etd.ohiolink.edu/rws_etd/document/get/ohiou1343799373/inline)
Figure 1: peninsularis (not “penisularis”) (there may be some other similar typos; I did not check everything)
l. 392: spinolai
l. 394-5: in his PhD thesis, JS Patterson reported observations and ELISA plus precipitin tests suggesting that Linshcosteus karupus are associated with wild canines, rodents, birds, and perhaps ungulates in their natural stony habitats in southern India; reptiles and bats were also observed, and the bugs were found to harbor the rodent trypanosome, T. conorrhini (Patterson 2007; see http://researchonline.lshtm.ac.uk/682370/). In a review on T. conorrhini, Dias and Campos Seabra (1943 Mem Inst Oswaldo Cruz 39:303-33) mention a precipitin study in which T. rubrofasciata appeared to have fed only on rats (p. 302-3; http://www.scielo.br/pdf/mioc/v39n3/tomo39(f3)_301-330.pdf).
l. 411: …this speciose group…
l. 452: Chagas
l. 494: “The Company of Biologists Ltd” [?]
l. 498: "SciELO Brasil" [?]
l. 505: referência (not “referenda”)
l. 516: Trypanosoma cruzi (italics)
l. 523 & 564: a repeated reference
l. 538-41: uppercase use
l. 546 & 557: another repeated reference
l. 548 & 554: yet another repeated reference
l. 577-9: uppercase use, journal name (and delete “journal”)
l. 600: species
S1 Table: geniculatus (not “geniculatis”)
S1 Table: Dias et al. 2010 did not study R. prolixus, just R. robustus

·

Basic reporting

The manuscript is well structured, with adequate language, figure and table.

Experimental design

The manuscript is according the scope of the journal. It presents two parts: in the first, a study with (1) triatomines captured and identification of the stomach contents by molecular method; the second, (2) works with literature records of host associations. The research question is relevant, but the sample is very little (1), and the results refer only to the knowledge available via web (2).

Validity of the findings

Despite the small sample of triatomines, the results seem to include new feeding sources not yet reported for triatomines (they don´t know about the older records) - and reinforce the improvement that the application of the molecular method for identification of food sources represents. Nevertheless, from the sample of 64 insects tested, only in 24 the food source was identified. That is, only insects with significant amount of blood in the stomach have the gut content identified, which represents a limitation of the technique. It is in accordance with other papers using similar methods.

Comments for the author

Title: . It limits the study to wild triatomines, but the manuscript also includes domestic and peridomestic insects.
Page 3, line 61: Rhodnius prolixus, not Rhodnius robustus. In contrary, the citation number 10 reports the unusual find of Rhodnius robustus associated to human. So, it is necessary to use other reference. I suggest this, because it is a reference publication: Lent H, Wygodzinsky P. (reference number 6)

Page 9, line 190: Blast
Page 11, line 218: What was the period surveyed? The availability of articles via internet is relatively recent in the history of the triatomine research, and has grown especially in the last 20 years. This way, previous publications cannot be found through electronic searches in databases. So, surely much information on the triatomines blood meal sources was not included in this research. The use of electronic search has great application, but these results must be relativized, because it would be unfair and wrong do not consider the previous knowledge.

Page 11, lines 238/239: The sample obtained, unfortunately, has low significance considering the extensive area to which it refers. What is the meaning of the results presented on a percentage on page 12? 50% from 20? 100% from 1? Etc...

Page 15, lines 293-296: Please, review the text.

---

## Round 0.2 · Minor Revisions

· Academic Editor

Minor Revisions

Both reviewers have much appreciated the improvements introduced in the manuscript and congratulate the authors for their work. Yet, one of them has underlined some remaining minor points that still need your attention.

Reviewer 1 ·

Basic reporting

See my comments below

Experimental design

No comment

Validity of the findings

See my comments below

Comments for the author

I think that the authors have done an excellent job, and also that the manuscript is now stronger. I only have a few minor suggestions.
l. 14: mainly in Central and South America
l. 26: how each record was collected
l. 30 and elsewhere: check use of the acronym U.S.
l. 69: although the word is a bit odd anyway (and its use could easily be avoided), I think “cleptohemophagy” should read “cleptohematophagy”
l. 240-1: delete one “for all matches”
l. 277-9: apart from this part being (as the authors note) quite speculative, please check number agreement (“females” then “her”)
l. 390-1: please emphatically indicate that that there is NO “vaccine” for T. cruzi, and that results in ref 66 are purely theoretical (and, I would add, completely unrealistic – who would support a wild predator vaccination campaign?). To enhance clarity, I’d also suggest to split this sentence into two – one about empirical evidence (with references) that predators are (no italics needed, I think) exposed to triatomines and another to comment on the modeling results by Stella et al. (incidentally, in an as yet not peer-reviewed manuscript)
l. 397-42: there seem to be some agreement problems here (“combining” with “offer”, or “each study” and “are”)
l. 404-5: same as above (“methods”… “offers”)

·

Basic reporting

Adequate according to suggestions

Experimental design

Adequate according to suggestions

Validity of the findings

Adequate according to suggestions

Comments for the author

I congratulate the authors for the quality of the data revision and of the text.
I recommend the publication of the manuscript.

---

## Round 0.3 · accepted · Accept

· Academic Editor

Accept

Thank you very much for the improvements introduced in the manuscript and congratulations for your nice work.